# Towards practical differentially private causal graph discovery

**Lun Wang**
University of California, Berkeley
wanglun@berkeley.edu

**Qi Pang**
Zhejiang University
pangqi@zju.edu.cn

**Dawn Song**
University of California, Berkeley
dawnsong@gmail.com

## Abstract

Causal graph discovery refers to the process of discovering causal relation graphs from purely observational data. Like other statistical data, a causal graph might leak sensitive information about participants in the dataset. In this paper, we present a differentially private causal graph discovery algorithm, `Priv-PC`, which improves both utility and running time compared to the state-of-the-art. The design of `Priv-PC` follows a novel paradigm called `sieve-and-examine` which uses a small amount of privacy budget to filter out "insignificant" queries, and leverages the remaining budget to obtain highly accurate answers for the "significant" queries. We also conducted the first sensitivity analysis for conditional independence tests including conditional Kendall's $\tau$ and conditional Spearman's $\rho$. We evaluated `Priv-PC` on 7 public datasets and compared with the state-of-the-art. The results show that `Priv-PC` achieves 10.61 to 293.87 times speedup and better utility. The implementation of `Priv-PC`, including the code used in our evaluation, is available at `https://github.com/sunblaze-ucb/Priv-PC-Differentially-Private-Causal-Graph-Discovery`.

## 1 Introduction

Causal graph discovery refers to the process of discovering causal relation graphs from purely observational data. Causal graph discovery has seen wide deployment in areas like genomics, ecology, epidemiology, space physics, clinical medicine, and neuroscience. The `PC` algorithm [30] is one of the most popular causal discovery algorithms. It is comprised of a series of independence tests like Spearman's $\rho$ [29], Kendall's $\tau$ [14], G-test [20] or $\chi^2$-test [21]. The algorithm starts by connecting all variables in the graph. If an independence test indicates that two variables are independent, the edge between the two variables will be removed from the causal graph. The process will continue until the edges between independent variables are totally removed.

Like other statistical data, a causal graph can leak information about participants in the dataset. For instance, Genome-Wide Association Studies involve finding causal relations between Single Nucleotide Polymorphisms (SNPs) and diseases. In this case, a causal link between a specific SNP and a disease may indicate the participation of a minority patient. However, the problem of *effective causal graph discovery with differential privacy* remains largely unsolved.

**State-of-the-art.** The most straightforward solution is to perturb all the independence tests in the `PC` algorithm with calibrated noise such as Laplace or Gaussian noise [9]. However, as pointed out in [33], the numerous independence tests incur *too much noise to output meaningful causal graphs*. Even tight composition techniques based on Rényi differential privacy [1, 22, 32] cannot address the issue. The state-of-the-art solution to differentially private causal graph discovery is `EM-PC` [33], a modification of the `PC` algorithm which uses the exponential mechanism to guarantee differential privacy. Instead of perturbing each independence test with noise, `EM-PC` randomly selects how many and which edges to delete using the exponential mechanism. In this way, `EM-PC` manages to achieve a relative balance between utility and privacy. However, `EM-PC` has two severe defects. First,

`EM-PC` suffers from *extremely slow computation* because: 1) many independence tests which should have been pruned have to be computed because the exponential mechanism can only deal with off-line queries; 2) the utility function used in applying the exponential mechanism is computationally intensive. In fact, the computation overhead of the utility score is so large that the implementation from the original paper [33] uses a greedy search to approximate the solution presented in the paper. It is unclear whether the differential privacy still holds given this compromise since the original sensitivity bound does not hold anymore. Second, `EM-PC` also suffers from low utility because it changes the intrinsic workflow of the `PC` algorithm. Concretely, `EM-PC` explicitly decides how many edges to delete while `PC` makes this decision in an on-line fashion. Thus, `EM-PC` does not converge to the `PC` algorithm and cannot achieve perfect accuracy even with substantial privacy budget.

**Proposed solution.** In this paper, we proposed `Priv-PC`, a differentially private causal graph discovery algorithm with *much less running time* and *better result utility* compared to `EM-PC`. The design of `Priv-PC` follows a novel paradigm called `sieve-and-examine`. Intuitively, `sieve-and-examine` spends a small amount of privacy budget to filter out "insignificant" queries and answers the rest of queries carefully with substantial privacy budget. The proof that `Priv-PC` is differentially private is straightforward. The challenge is to understand why it also gives less running time and better utility.

`Sieve-and-examine`, as the name indicates, comprises two sub-processes executing alternately: `sieve` and `examine`. In the context of causal graph discovery, the `sieve` process uses sub-sampled sparse vector technique [9, 2] to filter out variable pairs unlikely to be independent with a little privacy budget. Then the `examine` process uses Laplace mechanism [9] to carefully check the remaining variable pairs and decide whether they are really independent with substantial privacy budget.

We choose sparse vector technique for its nice properties. First, sparse vector technique can answer a large number of threshold queries but only pay privacy cost for those whose output is above the threshold[1]. Fortunately, in causal graph discovery, only a few independence tests will yield results above the threshold so with sparse vector technique, we can save much privacy cost. Second, sparse vector technique can deal with online queries, so redundant independence tests can be pruned adaptively once their target edge is removed due to a previous independence test. Thus, with sparse vector technique, we can get rid of the unnecessary independence tests in `EM-PC` and significantly accelerate private causal discovery. We propose to further accelerate the execution and reduce privacy cost by augmenting the sparse vector technique using sub-sampling without replacement [2].

However, sparse vector technique is known for its poor utility [19], which raises concern about the accuracy of `sieve-and-examine`. Actually, there exist two types of errors in `sieve-and-examine`. Type I error refers to mistakenly filtering out truly independent pairs. Type II error refers to the failure to filter out variable pairs that are not independent. To reduce the errors, we take a two-step approach. First, we suppress type I error by tweaking the threshold lower so the noise is more unlikely to flip over the output from independence to the opposite. The tweak, on the other hand, will increase the number of type II errors. Fortunately, type II errors can be corrected by the `examine` process with a high probability. Furthermore, the threshold tweak typically only increases type II errors slightly because a meaningful threshold should be far away from the clusters of both independent pairs and dependent pairs.

**Independence tests in Priv-PC.** The noise magnitude in `Priv-PC` grows proportionally to the sensitivity of the independence test (Section 2.1). Thus, to obtain an appropriate noise level, we conducted rigorous sensitivity analysis for commonly used conditional independence tests including conditional Kendall's $\tau$ [16, 31] and conditional Spearman's $\rho$ [31] (Appendix D, E). We finally chose Kendall's $\tau$ in `Priv-PC` because of its small sensitivity. It also remains an interesting open question how to integrate independence tests with infinite sensitivity such as G-test [20] or $\chi^2$-test [21] in `Priv-PC`.

## 2 Preliminaries

In this section, we review necessary background knowledge about differential privacy and causal graph discovery.

## 2.1 Differential Privacy

Differential privacy, formally introduced by Dwork et al. [8] has seen rapid development during the past decade and is accepted as the golden standard for private analysis.

**Definition 1** $((\epsilon, \delta)$-differential privacy). *A (randomized) algorithm $\mathcal{A}$ with input domain $D$ and output range $\mathcal{R}$ is $(\epsilon, \delta)$-differentially private if $\forall$ neighboring datasets $\mathcal{D}, \mathcal{D}' \in D$, and $\forall \mathcal{S} \subseteq \mathcal{R}$, we have that:*

$$\mathbb{P}[\mathcal{A}(\mathcal{D}) \in \mathcal{S}] \leq e^{\epsilon} \mathbb{P}[\mathcal{A}(\mathcal{D}') \in \mathcal{S}] + \delta$$

*If $\delta = 0$, it is called $\epsilon$-differential privacy or pure differential privacy.*

Intuitively, the definition requires a differentially private algorithm to produce similar outputs on similar inputs. A common approach to achieving differential privacy is to perturb the output with noise. The noise is carefully calibrated to appropriately mask the maximum difference of the output defined as sensitivity.

**Definition 2** ($\ell_k$-sensitivity). *The $\ell_k$-sensitivity of a function $f : D \to \mathcal{R}$ is:*

$$\Delta f = \max_{x, y \in \mathcal{D}, \|x - y\| = 1} \|f(x) - f(y)\|_k$$

Since all the independence tests in this paper output scalars, we omit the used norm and refer to the value as sensitivity uniformly.

Composability is an important property of differential privacy. If several mechanisms are differentially private, so is their composition. The privacy parameters of the composed mechanism can be derived using standard composition theorem like advanced composition [9] and moments accountant [1]. The sparse vector technique [9] can be viewed as a special case for composition because it can answer a large number of threshold queries while only paying privacy cost for queries above the threshold. We refer the interested readers to Appendix B for more details.

## 2.2 Causal Graph Discovery

In statistics, causal graphs are *directed acyclic graphs* (DAGs) used to encode assumptions about the data-generating process, which are formally defined as follows.

**Definition 3** (Causal Graph). *A causal graph $\mathcal{G}$ is a directed acyclic graph (DAG) represented by a vertex set $V = \{v_1, v_2, \cdots, v_k\}$ and an edge set $E \subseteq V \times V$. $Adj(\mathcal{G}, v_i)$ represents the adjacent set of node $v_i$ in graph $\mathcal{G}$. The skeleton of a DAG is the undirected version of the graph.*

Causal graph discovery refers to the process of discovering causal graphs under an observed distribution such as a dataset. The output of a causal graph discovery algorithm is a *completed, partially directed acyclic graph* (CPDAG) because the directions of some edges cannot be determined only based on the observational distribution.

There exist a variety of causal graph discovery algorithms and the `PC` algorithm is one of the most popular ones. The first step in the `PC` algorithm is to find the skeleton of the causal graph using conditional independence tests. Then the edges are directed based on some auxiliary information from the independence tests to obtain CPDAG. Because the second step does not touch the data, we only focus on the first step given the post-processing theorem [9] in differential privacy. The details of the `PC` algorithm is introduced in Section 3.2 and Appendix A.

## 2.3 Conditional Independence Test

Conditional independence test is an important building block in many causal discovery algorithms. It is used to test whether two random variables are independent conditional on another set of variables.

**Definition 4** (Conditional independence test). *A conditional independence test $f : V \times V \times 2^V \times D \to \{0, 1\}$ decides whether variable $i \neq j \in V$ are independent conditional on another set of variables $k \subseteq V, i, j \notin k$. $f$ is composed of a dependence score $s : V \times V \times 2^V \times D \to \mathbb{R}$ and a threshold $T \in \mathbb{R}$.*

$$f(\mathcal{D}) = \begin{cases} 0, & s(\mathcal{D}) \leq T \\ 1, & s(\mathcal{D}) > T \end{cases}$$

*, where $1$ represents "independent" and $0$ represents "not independent". $f$ is called $|k|$-order conditional independence test where $|k|$ is the size of the conditional set.*

Commonly used independence tests include Spearman's $\rho$, Kendall's $\tau$, G-test and $\chi^2$-test. Note that some independence tests like Kendall's $\tau$ output 1 when the dependence score is below the threshold. However, for clarity, we assume all the independence tests output 1 when the dependence score is above the threshold without loss of generality. In this paper, we focus on Kendall's $\tau$ because of its small sensitivity (Section 3.3).

## 3 Differentially Private Causal Graph Discovery

In this section, we proposed `Priv-PC` to effectively discover causal graphs following `sieve-and-examine` paradigm. Concretely, `Priv-PC` leverages the `sieve` process to sift out variable pairs unlikely to independent using a little privacy cost and then carefully `examines` the remaining ones with substantial privacy budget. We first introduce `sieve-and-examine` mechanism and then demonstrate how to apply `sieve-and-examine` to the PC algorithm to obtain `Priv-PC`. At last, we bridge `sieve-and-examine` and `Priv-PC` by providing sensitivity analysis for Kendall's $\tau$.

### 3.1 Sieve-and-examine Mechanism

Most causal graph discovery algorithms like the PC algorithm need to answer many independence tests – *too many to obtain an acceptable privacy guarantee* using independent perturbation mechanisms like Laplace mechanism [9]. `EM-PC` is the first step towards reconciling the contradiction between utility and privacy in private causal discovery. However, `EM-PC` suffers from *extremely slow running time* because it additionally runs a large number of independence tests that should have been pruned. A straightforward solution is to replace the exponential mechanism [9] with the sparse vector technique [9, 19]. Sparse vector technique allows adaptive queries so unnecessary independence tests can be pruned early. Besides, the privacy cost of sparse vector technique only degrades with the number of queries above the threshold. Fortunately, only a few independence tests in causal discovery yield values above the threshold so the sparse vector technique can also save considerable privacy budget in causal discovery. However, sparse vector technique suffers from *low accuracy* as pointed out in [19], which is not acceptable in many use cases such as medical or financial analysis.

To address the issue, we propose a novel paradigm called `sieve-and-examine` which alternately executes sub-sampled sparse vector technique and output perturbation. Intuitively, the `sieve` process uses sub-sampled sparse vector technique to filter out independence tests unlikely to be above the threshold with small privacy budget. Then the left queries are `examined` carefully with substantial privacy budget using output perturbation.

**One-off sieve-and-examine.** For simplicity, we first introduce one-off `sieve-and-examine` shown in Algorithm 1, a simplified version of `sieve-and-examine` that halts after seeing one query above the threshold. We prove that one-off `sieve-and-examine` is $\epsilon$-differentially private. The result can be generalized to multiple above-threshold queries using composition theorem.

---

**Algorithm 1:** One-off sieve-and-examine mechanism.

**Input:** $\mathcal{D}$: dataset, $\{f_i\}$: queries, $T$: threshold, $t$: threshold tweak, $m$: subset size, $\epsilon$: privacy parameters, $\Delta$: sensitivity of $f$

1 . **Function** *Sieve_and_examine($\mathcal{D}, \{f_i\}, T, t, m, \epsilon, \Delta$)*:
2      $\mathcal{D}' \xleftarrow{\$} \mathcal{D}, n = |\mathcal{D}|, m = |\mathcal{D}'|$;
3      Let $\epsilon' = \ln(\frac{n}{m}(e^{\epsilon/2} - 1) + 1)$;
4      Let $\hat{T} = T - t + Lap(\frac{2\Delta}{\epsilon'})$;
5      **for** *Each query $i$* **do**
6          **if** $f_i(\mathcal{D}') + Lap(\frac{4\Delta}{\epsilon'}) \geq \hat{T}$ **then**
7              Let $k = i$;
8              Break;
9      **if** $f_k(\mathcal{D}) + Lap(\frac{2\Delta}{\epsilon}) \geq T$ **then** Output k ;
10      **else** Output $\perp$;

---

**Theorem 1.** *Algorithm 1 is $\epsilon$-differentially private.*

*Proof Sketch.* We separately prove that `sieve` and `examine` are both $\epsilon/2$-differentially private. The main body of `sieve` is a sparse vector technique with $\epsilon' = \ln(\frac{n}{m}(e^{\epsilon/2} - 1) + 1)$ privacy cost. Sub-sampling reduces the cost to $\epsilon/2$ following Theorem 9 from [2]. `Examine` process is a $\epsilon/2$-differentially private Laplace mechanism. Thus, `sieve-and-examine` is $\epsilon$-differentially private using composition theorem. □

**Result Utility.** The differential privacy proof is straightforward. The challenge will be to understand when it also gives utility. Thus, we bound the probability of type I error and type II error in Algorithm 1 separately and provide the proof in Appendix C.

**Theorem 2** (Error bound)**.**

- *(Type I error) Let $E_1^\alpha$ denotes the event that Algorithm 1 filters out $f(\mathcal{D}) \geq T + \alpha$.*

$$\mathbb{P}[E_1^\alpha] \leq \exp(-\frac{\epsilon'(\alpha + t)}{6\Delta}) - \frac{1}{4}\exp(-\frac{\epsilon'(\alpha + t)}{3\Delta})$$

- *(Type II error) Let $E_2^\alpha$ denotes the event that Algorithm 1 fails to filter out $f(\mathcal{D}) \leq T - \alpha$. If $\alpha \geq t$, then*

$$\mathbb{P}[E_2^\alpha] \leq \exp(-\frac{12\epsilon\alpha + \epsilon'(\alpha - t)}{6\Delta}) - \frac{1}{4}\exp(-\frac{6\epsilon\alpha + \epsilon'(\alpha - t)}{3\Delta})$$

.

Intuitively, theorem 2 bounds the probability of errors conditional on the distance from the dependence score to the threshold. An interesting observation is the tweak on the threshold $t$ decreases the probability of type I errors and increases the probability of type II errors at the same time. Because each type II error increases the privacy cost by $\epsilon$, the question is "*will the increment of type II errors add too much privacy cost?*" Fortunately, the answer is "no" because the increment of type II errors also depends on the distribution of dependence scores. Generally the empirical distribution of an independence score is a twin-peak curve and the threshold locates in the middle valley. In this case, the threshold tweak only slightly increases the number of type II errors because most dependence scores are far from the threshold[2].

## 3.2 Priv-PC Algorithm

In this section, we demonstrate how to apply `sieve-and-examine` to PC algorithm to obtain `Priv-PC`. We first give an overview of `Priv-PC`. Then we discuss how to optimize the sub-sampling rate in `Priv-PC`.

`Priv-PC` **algorithm.** The complete pseudo-code for `Priv-PC` is shown in Algorithm 2. `Priv-PC` follows the same workflow as the PC algorithm. It starts from a complete undirected graph (line 1) and gradually increases the order of the independence tests (line 6, 17). Within a fixed order, `Priv-PC` traverse all the variable pairs with large enough adjacent set (line 8). It selects the conditional variables from the adjacent set (line 9-10) and then executes the conditional independence test to decide whether the edge will be removed from the graph.

To achieve differential privacy, the conditional independence tests are augmented with `sieve-and-examine`. Concretely, `Priv-PC` first sub-samples a subset $\mathcal{D}'$ from $\mathcal{D}$, derives privacy parameter for the `sieve` process and tweaks the threshold (line 3-5). Then, `Priv-PC` executes the `sieve` process by adding noise to both the tweaked threshold (line 5) and the independence test (line 11). Note that the noise parameters here are different from standard `sieve-and-examine` (Algorithm 1) because the sensitivity for Kendall's $\tau$ is dependent on the dataset size (Section 3.3). Once an independence test on the sub-sampled dataset exceeds the threshold (line 11), the `examine` process will run the independence test again on the complete dataset with substantial privacy budget. If the result still exceeds the un-tweaked threshold (line 12), the edge is removed from the graph (line 13). Then, the sub-sampled dataset and the threshold are refreshed for the next round of `sieve-and-examine` (line 14-15).

.

**Algorithm 2:** `Priv-PC` Algorithm with Kendall's $\tau$. The highlighted parts are different from `PC` algorithm.

**Input:** $V$: vertex set, $\mathcal{D}$: dataset, $T$: threshold, $t$: threshold tweak, $m$: subset size, $\epsilon$: privacy parameter, $\Delta$: sensitivity on the full dataset.

1 **Function** *Priv_PC(V, $\mathcal{D}$, T, t, m, $\epsilon$, $\Delta$)*:
2     $\mathcal{G}$ = complete graph on V, $ord = 0$
3     $\mathcal{D}' \xleftarrow{\$} \mathcal{D}, n = |\mathcal{D}|, m = |\mathcal{D}'|$
4     Let $\epsilon' = \ln(\frac{n}{m}(e^{\epsilon/2} - 1) + 1)$
5     Let $\hat{T} = T - t + Lap(\frac{2\sqrt{n}\Delta}{\sqrt{m}\epsilon'})$
6     **while** $\exists\, v_i$ *s.t.* $|Adj(\mathcal{G}, v_i) - v_j| \geq ord$ **do**
7        **while** $\exists$ *edge* $(v_i, v_j)$ *s.t.* $|Adj(\mathcal{G}, v_i) - v_j| \geq ord$ *that has not been tested* **do**
8           select edge $(v_i, v_j)$ in $\mathcal{G}$ s.t. $|Adj(\mathcal{G}, v_i) - v_j| \geq ord$
9           **while** $\exists\, S \subseteq Adj(\mathcal{G}, v_i) - v_j$ *that has not been tested* **do**
10             choose $S \subseteq Adj(\mathcal{G}, v_i) - v_j, |S| = ord$
11             **if** $\tau(ij|S) + Lap(\frac{4\sqrt{n}\Delta}{\sqrt{m}\epsilon'}) \geq \hat{T}$ **then**
12                **if** $\tau(ij|S) + Lap(\frac{2\Delta}{\epsilon}) \geq T$ **then**
13                   delete $(v_i, v_j)$ from $\mathcal{G}$
14                $\mathcal{D}' \xleftarrow{\$} \mathcal{D}, |\mathcal{D}'| = m$
15                $\hat{T} = T - t + Lap(\frac{2\sqrt{n}\Delta}{\sqrt{m}\epsilon'})$
16                break
17     $ord = ord + 1$
18     Output $\mathcal{G}$, compute the total privacy cost $(\epsilon_{tot}, \delta_{tot})$ with advanced composition.

**Optimize sub-sampling rate in** `Priv-PC`**.** In Algorithm 2, we require the caller of the function to explicitly give the size of the sub-sampling set. However, since the sensitivity of Kendall's $\tau$ also depends on the data size (Section 3.3), we can actually derive an optimal sub-sampling size which adds the smallest noise under the same privacy guarantee. This requires to minimize the noise level $\frac{\sqrt{n/m}}{\ln(\frac{n}{m}(\exp(\epsilon/2)-1)+1)}$. Although there is no explicit solution for the optimization problem, we can obtain an approximate solution with numerical solver such as BFGS [24]. On the other hand, when $\epsilon$ is small, the optimal sub-sampling size is also too small to yield meaningful independence test results. Thus we take the optimal sub-sampling size by clipping the solution to range $\left(\frac{n}{20}, n\right)$.

### 3.3 Independence Tests in Priv-PC

The last missing piece is the sensitivity of the conditional independence test functions. We finally choose conditional Kendall's $\tau$ for its small sensitivity. Conditional Spearman's $\rho$ is another candidate but it can only be used on large datasets because of the large coefficient in its sensitivity (Appendix E).

The sensitivity of Kendall's $\tau$ is inversely proportional[3] to the training set size as pointed out in [17]. However, in our scenario, the conditional version of Kendall's $\tau$ is needed while [17] only gives the sensitivity for non-conditional Kendall's $\tau$. In order to fill the gap, we derive the sensitivity of the conditional Kendall's $\tau$, and leave the proof to Appendix D.

**Theorem 3.** *(Sensitivity of conditional Kendall's $\tau$.) The sensitivity of conditional Kendall's $\tau$ in Definition 6 (Appendix D) is $\frac{c_1}{\sqrt{n}}$ where $n$ is the size of the input dataset and $c_1$ is an explicit constant approaching $9/2$ when the dataset size grows.*

## 4 Evaluation

In this section, we evaluate the effectiveness of `Priv-PC` by answering the following two questions. 1) How accurate is the result of `Priv-PC`? 2) How much running time does `Priv-PC` save?

## 4.1 Experiment Setup

In order to answer the above questions, we selected 7 datasets. The detailed information about the datasets is shown in Table 1.

Table 1: Datasets used in the evaluation.

| Dataset | # Features | # Samples | # Edges | Type |
|---|---|---|---|---|
| Earthquake [15] | 5 | 100K | 4 | Binary |
| Cancer [15] | 5 | 100K | 4 | Binary |
| Asia [18] | 8 | 100K | 10 | Binary |
| Survey [27] | 6 | 100K | 6 | Discrete |
| Alarm [3] | 37 | 100K | 46 | Discrete |
| Sachs [26] | 11 | 100K | 17 | Discrete |
| Child [4] | 20 | 100K | 25 | Discrete |

To directly compare `EM-PC` and `Priv-PC`, we ran the two algorithms on the datasets with 21 different privacy parameters and presented the results with accumulated privacy cost between 1 and 100. Furthermore, to demonstrate the utility improvement due to `sieve-and-examine`, we also directly applied sparse vector technique to `PC` algorithm (`SVT-PC`) and evaluated it under the same setting. For each privacy parameter, we ran the three algorithms for 5 times and recorded the mean and standard deviation of the utility of the output graph and the running time. We fix $\delta = 1e\text{-}3$ for both `EM-PC` and `Priv-PC` across all the experiments. Utility is measured in terms of $F1$-score[4]. All the experiments were run on a Ubuntu18.04 LTS server with 32 AMD Opteron(TM) Processor 6212 with 512GB RAM.

## 4.2 Utility

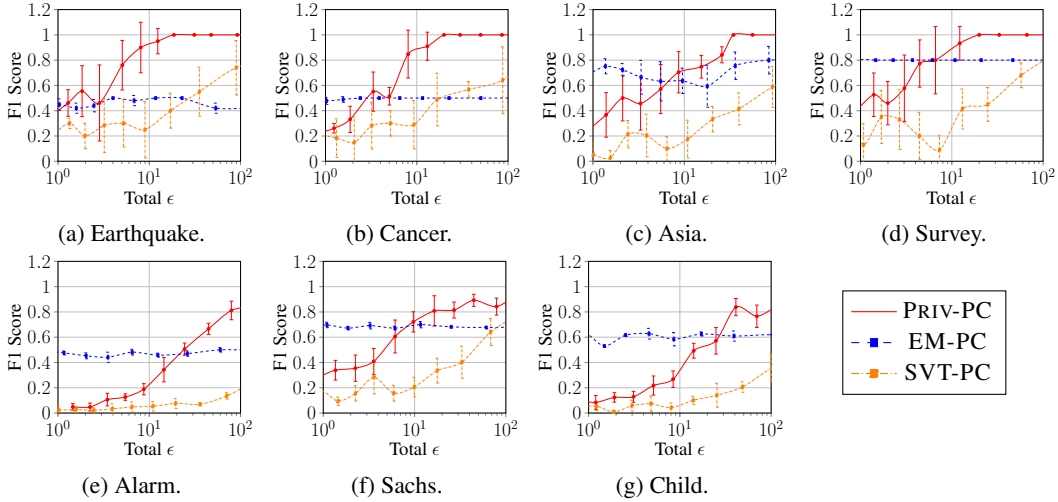

(a) Earthquake.  (b) Cancer.  (c) Asia.  (d) Survey.

(e) Alarm.  (f) Sachs.  (g) Child.

Figure 1: $F1$-Score vs. Privacy Budget.

In the evaluation, `Priv-PC` achieves better utility than `EM-PC` when the privacy budget is reasonably large as shown in Figure 1. `Priv-PC` always converges to perfect accuracy when privacy cost grows while `EM-PC` does not. The reason is that `Priv-PC` converges to `PC` when privacy cost grows but `EM-PC` does not because it contains a unique sub-routine to explicitly decide the number of edges to delete. The sub-routine intrinsically inhibits the accuracy of `EM-PC`. On the other hand, `EM-PC` achieves better utility under small privacy budget[5] because the exponential mechanism has better utility than the sparse vector technique under small privacy budget as pointed out in [19].

Compared with `SVT-PC`, `Priv-PC` always achieves much better utility in the medium privacy region (Figure 1). The improvement should be attributed to `sieve-and-examine` because it effectively suppresses type I and type II errors in sparse vector technique 3.1. Second, because the sensitivity of Kendall's $\tau$ is inversely proportional to the size of the input dataset, the noise is typically small when the dataset is large. Thus, the noise does not severely harm the utility while preserving rigorous privacy.

### 4.3 Running Time

| Average Running Time | EM-PC | SVT | Priv-PC | Priv-PC w/o sub-sampling |
|---|---|---|---|---|
| Earthquake [15] | 176.04s | 3.38s | 6.62s | 11.01s |
| Cancer [15] | 64.62s | 2.94s | 6.09s | 10.83s |
| Asia [18] | 531.80s | 10.06s | 16.19s | 19.40s |
| Survey [27] | 68.13s | 1.21s | 2.13s | 5.12s |
| Alarm [3] | 10601.33s | 71.23s | 143.01s | 315.32s |
| Sachs [26] | 4858.42s | 4.29s | 16.65s | 72.98s |
| Child [4] | 25140.67s | 32.85s | 85.55s | 191.31s |

Table 2: Running time when privacy budget for each `sieve-and-examine` is 1.

`Priv-PC` achieves 10.61 to 32.85 times speedup on small graphs and 74.13 to 293.87 times speedup on larger graphs compared with `EM-PC` as shown in Table 2. The improvement is due to two reasons. First, `Priv-PC` can deal with online queries while `EM-PC` cannot. Thus, if an edge is removed due to a previous independence test, later tests on the same edge can be skipped to avoid extra computation overhead. Second, in the `sieve` process, `Priv-PC` only runs independence tests on a subset of the dataset which further accelerates the process. This also explains why `Priv-PC` sometimes runs faster than `SVT-PC`.

| #IDP tests | Priv-PC | EM-PC |
|---|---|---|
| Asia | 95 | 216 |
| Cancer | 37 | 57 |
| Earthquake | 40 | 61 |
| Survey | 29 | 38 |
| Alarm | 1843 | 12979 |
| Sachs | 165 | 1224 |
| Child | 1162 | 7393 |

Table 3: The number of independence tests in `Priv-PC` and `EM-PC`.

To better understand how the two factors contribute to the speedup, we run `Priv-PC` without sub-sampling under the same setting and include the results in Table 2. The results show that on small graphs, the first factor provides 5.97 to 27.41 times speedup and sub-sampling provides 1.20 to 2.40 times speedup; on larger graphs, the first factor provides 33.62 to 131.41 times speedup and sub-sampling provides 2.20 to 4.38 times speedup.

To better illustrate the source of the speedup, we measure the number of independence tests conducted in `EM-PC` and `Priv-PC` as shown in Table 3. The results show that `Priv-PC` saves 34.4% to 56.0% independence tests on small graphs and 84.3% to 86.8% on larger graphs compared to `EM-PC`.

## 5   Related Work

Causal inference has a long history and there are several excellent overviews [25, 10] of this area. In this section, we briefly introduce the related works in the two most relevant sub-areas: causal discovery based on graph models and private causal inference.

Causal discovery based on graph models can be roughly classified into two categories. The first category is constraint-based causal discovery. The `PC` algorithm [30] is the most well-known algorithm in this category. It traverses all the edges and adjacent conditional sets in the causal graph and removes the edge if the conditional independence test indicates that the edge connects two independent variables. An important variation of the `PC` algorithm is the Fast Causal Inference (FCI) [30], which tolerates latent confounders. The Greedy Equivalence Search (GES) [6] is another widely-used algorithm in this category which starts with an empty graph and gradually adds edges. The second category is based on functional causal models (FCM). A FCM represents the effect Y as a function of the direct causes X and some noise: $Y = f(X, \epsilon; \theta)$. In Linear, Non-Gaussian and Acyclic Model (LiNGAM) [28], $f$ is linear, and only one of $\epsilon$ and $X$ can be Gaussian. In Post-Nonlinear Model (PNL) [34, 35], $Y = f_2(f_1(X) + \epsilon)$. Additive Noise Model (ANM) [12] further constrains the post-nonlinear transformation of PNL model.

Private causal discovery has a relatively short history. In 2013, Johnson *et al.* [13] studied differentially private Genome-Wide Association Studies (GWAS). They used Laplace mechanism and exponential mechanism to build specific queries of interest in GWAS. In 2015, Kusner *et al.* [17] analyzed the sensitivity of several commonly used dependence scores on training and testing datasets, and then applied Laplace mechanism to the ANM model. Xu *et al.* [33] proposed to apply exponential mechanism to the PC algorithm. Another line of work focuses on private bayesian inference including [7, 11, 5]. Their pioneering works are inspiring but lack novelty from differential privacy side because they all directly leverage off-the-shelf differentially private mechanisms without any modification.

## 6 Conclusion & Future Work

This paper takes an important step towards practical differentially private causal discovery. We presented Priv-PC, a novel differentially private causal discovery algorithm with high accuracy and short running time. We also performed an empirical study to demonstrate the advantages compared with the state-of-the-art.

At the same time, we observe many challenges in differentially private causal discovery that existing techniques are not capable of handling. For example, it is unclear how to reconcile independence tests with infinite sensitivity such as G-test and $\chi^2$-test; it is unclear how to handle data type beyond categorical data like numerical data since PC algorithm only handles discrete data. We consider all these problems as important future work in the research agenda toward solving the private causal discovery problem.

## Potential Broader Impact

Priv-PC provides an approach to effectively discovering causal graphs from purely observational data. It can be deployed in genomics, ecology, epidemiology, space physics, clinical medicine, and neuroscience to release causal graph while preserving the privacy of the sensitive input data. At the same time, Priv-PC should be used with carefully calibrated parameters to make sure the discovered graph is accurate and preserves the privacy . Inappropriate parameters might lead to weak privacy guarantee that does not provide strong protection against attacks such as inference attack.

## Acknowledgments and Disclosure of Funding

The authors would like thank Joseph P. Near and Chenguang Wang for their valuable advice on the manuscript, and the anonymous reviewers for their helpful comments. This material is in part based upon work supported by DARPA contract N66001-15-C-4066, the Center for Long-Term Cybersecurity, and Berkeley Deep Drive. Any opinions, findings, conclusions, or recommendations expressed in this material are those of the authors, and do not necessarily reflect the views of the sponsors.

## Footnotes

[1]Sparse vector technique can also only pay for queries below the threshold. For clarity, we only focus on the above-threshold queries throughout the paper.

[2]A complete explanation contains two parts. First, since most of the dependence scores are far from the threshold, the threshold tweak does not directly change the test results for most queries. Second, because dependence scores are far from the threshold, the absolute increase of type II error probability is small. Thus, the increment of type II errors is small.

[3]Note that this requires the size of the dataset to be public which is a common case.

[4]If a causal graph discovery outputs $\mathcal{G} = (V, E)$ and the ground truth is $\mathcal{G}' = (V, E')$. Then $F1$-score is defined as: $F1 = \frac{2 \cdot \text{Precision} \cdot \text{Recall}}{\text{Precision} + \text{Recall}}$, $\text{Precision} = \frac{|E \cap E'|}{|E|}$, $\text{Recall} = \frac{|E \cap E'|}{|E'|}$

[5]The line for `EM-PC` is almost flat in Figure 1 because the rising segment appears under small privacy budget out of the axis scope (approximately $0.01 \sim 0.1$ according to our evaluation).

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
