[Supplementary Material]

## A  Pseudo-code of PC Algorithm

For completeness, we provide the pseudo-code for the `PC` algorithm in Algorithm 3.

---
**Algorithm 3:** `PC` Algorithm.

---
**Input:** $V$: vertex set, $\mathcal{D}$: dataset, $T$: threshold
1 **Function** *PC($V, \mathcal{D}, T$)*:
2     $\mathcal{G}$ = complete graph on V, *ord* = 0
3     **while** $\exists\, v_i\ s.t.\ |Adj(\mathcal{G}, v_i) - v_j| \geq ord$ **do**
4        **while** $\exists\ edge\ (v_i, v_j)\ s.t.\ |Adj(\mathcal{G}, v_i) - v_j| \geq ord\ that\ has\ not\ been\ tested$ **do**
5           select edge $(v_i, v_j)$ in $\mathcal{G}$ $s.t.\ |Adj(\mathcal{G}, v_i) - v_j| \geq ord$
6           **while** $\exists\ S\ that\ has\ not\ been\ tested$ **do**
7              choose $S \subseteq Adj(\mathcal{G}, v_i) - v_j, |S| = ord$
8              **if** $indep\_test(ij|S) \geq T$ **then**
9                 remove $(v_i, v_j)$ from $\mathcal{G}$
10                 break
11     *ord* = *ord* + 1
12     Output $\mathcal{G}$

---

## B  Exponential Mechanism & Sparse Vector Technique

For completeness, we provide detailed description of exponential mechanism and exponential mechanism.

**Exponential Mechanism.**  Exponential mechanism is designed for differentially private selection from infinite output set. It computes an utility score for each candidate output and randomly selects from the output candidates based on probability derived from the utility score. The pseudo-code for exponential mechanism is shown in Algorithm 4.

---
**Algorithm 4:** Exponential Mechanism

---
**Input:** $\mathcal{D}$: dataset, $O$: output set, $u$: 1-sensitive utility function, $\epsilon$: privacy parameters.
1 **Function** *EM($\mathcal{D}, O, u, \epsilon$)*:
2     Initiate $U$ as an empty lists
3     **for** $o \in O$ **do**
4        Append $u(\mathcal{D}, o)$ to $U$
5     Randomly select $o$ from $O$ according to probability $\frac{\exp(\epsilon u_o/2)}{\sum_{u_i \in U} \exp(\epsilon u_i/2)}$.

---

**Sparse Vector Technique.**  Sparse vector technique is a widely used differentially private mechanism. It can answer a large number of queries while only paying privacy cost for a small portion of them. The pseudo-code for sparse vector technique is shown in Algorithm 5.

## C  Proof for Error Bound

**Theorem 4** (Type I error bound). *Let $E_1^\alpha$ denotes the event that Algorithm 1 filters out $f(D) \geq T + \alpha$.*

$$\mathbb{P}[E_1^\alpha] \leq \exp(-\frac{\epsilon'(\alpha + t)}{6\Delta}) - \frac{1}{4}\exp(-\frac{\epsilon'(\alpha + t)}{3\Delta})$$

.

*Proof.* We want to upper bound the probability of $E_1^\alpha$. Equally, we lower bound the probability of $\neg E_1^\alpha$ by the probability that the noise on the threshold is smaller than $\frac{1}{3}(t + \alpha)$ and the noise on the

---

**Algorithm 5:** Sparse Vector Technique.

---

**Input:** $D$: dataset, $\{f_i\}$: 1-sensitive queries, $T$: threshold,
$\quad\quad c$: quota of above-threshold queries, $(\epsilon, \delta)$: privacy parameters.

**1 Function** *SVT($\mathcal{D}, \{f_i\}, T, c, \epsilon, \delta$)*:

**2** $\quad$ **if** $\delta = 0$ **then** Let $\sigma = \frac{2c}{\epsilon}$ **else** Let $\sigma = \frac{\sqrt{32c \log \frac{1}{\delta}}}{\epsilon}$

**3** $\quad$ Let count = 0, $\hat{T}_{count} = T + Lap(\sigma)$

**4** $\quad$ **for** *Each query $i$* **do**

**5** $\quad\quad$ Let $\nu = Lap(2\sigma)$

**6** $\quad\quad$ **if** $f_i(\mathcal{D}) + \nu_i \geq \hat{T}_{count}$ **then**

**7** $\quad\quad\quad$ Output $a_i = \top$

**8** $\quad\quad\quad$ Let count = count + 1 Let $\hat{T}_{count} = T + Lap(\sigma)$

**9** $\quad\quad$ **else** Output $a_i = \bot$

**10** $\quad\quad$ **if** *count $\geq q$* **then** Halt

---

query output is smaller than $\frac{2}{3}(t + \alpha)$. Because for Laplace noise, $\mathbb{P}[x \geq w] = \exp(-w/b)$, we have

$$\mathbb{P}[\neg E_1^\alpha] \geq (1 - \frac{1}{2}\exp(-\frac{\epsilon'(\alpha + t)}{6\Delta}))^2 = 1 - \exp(-\frac{\epsilon'(\alpha + t)}{6\Delta}) + \frac{1}{4}\exp(-\frac{\epsilon'(\alpha + t)}{3\Delta})$$

. Thus,

$$\mathbb{P}[E_1^\alpha] \leq 1 - \mathbb{P}[\neg E_1^\alpha] \leq \exp(-\frac{\epsilon'(\alpha + t)}{6\Delta}) - \frac{1}{4}\exp(-\frac{\epsilon'(\alpha + t)}{3\Delta})$$

408 $\hfill\square$

**Theorem 5** (Type II error bound). *Let $E_2^\alpha$ denotes the event that Algorithm 1 fails to filter out $f(D) \leq T - \alpha$. If $\alpha \geq t$, then*

$$\mathbb{P}[E_2^\alpha] \leq \exp(-\frac{12\epsilon\alpha + \epsilon'(\alpha - t)}{6\Delta}) - \frac{1}{4}\exp(-\frac{6\epsilon\alpha + \epsilon'(\alpha - t)}{3\Delta})$$

409 .

*Proof.* If $f(D)$ is not filtered out, it needs to be missed by both sparse vector technique and the Laplace mechanism. The probability bound for being missed by the sparse vector technique is

$$\mathbb{P}[E_{svt}^\alpha] \leq \exp(-\frac{\epsilon'(\alpha - t)}{6\Delta}) - \frac{1}{4}\exp(-\frac{\epsilon'(\alpha - t)}{3\Delta})$$

following similar proof path to theorem 4. The probability being missed by the Laplace mechanism is bounded by

$$\mathbb{P}[E_{lm}^\alpha] = \exp(-\frac{2\epsilon\alpha}{\Delta})$$

410 . Thus,

$$\mathbb{P}[E_2^\alpha] = \mathbb{P}[E_{svt}^\alpha] \cdot \mathbb{P}[E_{lm}^\alpha] \leq (\exp(-\frac{\epsilon'(\alpha - t)}{6\Delta}) - \frac{1}{4}\exp(-\frac{\epsilon'(\alpha - t)}{3\Delta})) \cdot \exp(-\frac{2\epsilon\alpha}{\Delta})$$

$$= \exp(-\frac{12\epsilon\alpha + \epsilon'(\alpha - t)}{6\Delta}) - \frac{1}{4}\exp(-\frac{6\epsilon\alpha + \epsilon'(\alpha - t)}{3\Delta})$$

411 $\hfill\square$

## 412 D Sensitivity of Kendall's $\tau$

413 In this section, we derive the sensitivity of Kendall's $\tau$ and its conditional version. We first give the
414 complete definition of Kendall's $\tau$ and its conditional version.

**Definition 5** (Kendall's $\tau$). *Let* $\{(a_1, b_1), \cdots, (a_n, b_n)\}$ *denotes the observations. A pair of observation indices* $(i, j)$ *are called* concordant *if* $a_i > a_j$ *and* $b_i > b_j$. *Otherwise* $(i, j)$ *is called* discordant. *Kendall's* $\tau$ *is defined as*

$$\tau_{ij} := \frac{2|C - D|}{n(n-1)}$$

415 *where* $C$ *is the number of concordant pairs and* $D$ *is the number of discordant pairs.*

416 Kusner et al. [15] derive the sensitivity for unconditional Kendall's $\tau$ when the neighboring relation
417 between datasets are constrained to replacement. The first step towards complete sensitivity analysis
418 for unconditional Kendall's $\tau$ is to extend the neighboring relation to increment.

419 **Theorem 6.** *Kendall's* $\tau$ *is* $\frac{2}{n-1}$-*sensitive.*

420 *Proof.* When the neighboring datasets are defined by replacement, the proof is done in [15]. Now
421 we prove that the sensitivity bound generalizes to neighboring datasets defined by increment.

422 If we increment a dataset by one row, $|C - D|$ can increase by at most $n$.

$$s(\tau_{ij}) \leq \frac{|C - D| + n}{\frac{1}{2}n(n+1)} - \frac{|C - D|}{\frac{1}{2}n(n-1)} \leq \frac{|C - D| + n}{\frac{1}{2}n(n-1)} - \frac{|C - D|}{\frac{1}{2}n(n-1)} \leq \frac{2}{n-1}$$

423 □

424 **Theorem 7.** *If the conditional variables have* $k$ *blocks, then conditional Kendall's* $\tau$ *is* $\frac{c_\tau}{\sqrt{n-1}}$-
425 *sensitive, where* $c_\tau$ *is an explicit constant typically close to* $\frac{9}{2}$.

**Definition 6** (Conditional Kendall's $\tau$). *We omit the pair indices* $i, j$ *and use* $\tau_i$ *to represent Kendall's* $\tau$ *in the* $i$th *block of the conditional variables. If there are* $k$ *blocks in total, then conditional Kendall's* $\tau$ *is defined as*

$$\tau = \frac{\sum_{i=1}^{k} w_i \tau_i}{\sqrt{\sum_{j=1}^{k} w_j}}$$

426 *where* $w_i = \frac{9n_i(n_i - 1)}{2(2n_i + 5)}$ *is the inverse of* $\tau_i$'s *variance.*

427 *Proof.* If the $i$th block contains $n_i$ observations, then $s(\tau_i) = \frac{2}{n_i - 1}$.

Then we need to bound $\frac{w_i}{\sqrt{\sum_{j=1}^{k} w_j}}$ and its sensitivity. Assuming $\forall i \in [1, k], n_i \geq c_1$, then $c_2(n_i - 1) \leq w_i \leq \frac{9(n_i - 1)}{4}$ for some explicit constants $c_2 = \frac{9c_1}{2(2c_1 + 5)}$. Thus

$$\frac{w_i}{\sqrt{\sum_{j=1}^{k} w_j}} \leq \frac{9(n_i - 1)}{4\sqrt{c_2(n - k)}}$$

and

$$s\left(\frac{w_i}{\sqrt{\sum_{j=1}^{k} w_j}}\right) \leq \frac{w_i'}{\sqrt{\sum_{j \neq i} w_j + w_i'}} - \frac{w_i}{\sqrt{\sum_{j=1}^{k} w_j}} \leq \frac{w_i' - w_i}{\sqrt{\sum_{j=1}^{k} w_j}} \leq \frac{9}{4\sqrt{c_2(n - k)}}$$

. Thus the complete sensitivity is bounded as follow.

$$s(\tau) \leq \left(\frac{w_i}{\sqrt{\sum_{j=1}^{k} w_j}} + s\left(\frac{w_i}{\sqrt{\sum_{j=1}^{k} w_j}}\right)\right)(\tau_i + s(\tau_i)) - \frac{w_i}{\sqrt{\sum_{j=1}^{k} w_j}}\tau_i \leq \frac{27}{4\sqrt{c_2(n - k)}} + \frac{9}{2c_1\sqrt{c_2(n - k)}}$$

428 □

 # E   Sensitivity of Spearman's $\rho$

In this section, we derive the sensitivity of Spearman's $\rho$ and its conditional version. We first give the complete definition of Spearman's $\rho$ and its conditional version.

**Definition 7** (Spearman's $\rho$). *Let $\{(a_1, b_1), \cdots, (a_n, b_n)\}$ denotes the observations. If we independently sort the observations $\{a_1, \cdots, a_n\}$ and $\{b_1, \cdots, b_n\}$ in ascending order. Let $d_i$ represent the distance between the order of $a_i$ and $b_i$. Spearman's $\rho$ is defined as*

$$\rho = |1 - \frac{6\sum_{i=1}^{n} d_i^2}{n(n^-1)}|$$

.

Kusner et al. [15] derive the sensitivity for unconditional Spearman's $\rho$ when the neighboring relation between datasets are constrained to replacement. The first step towards complete sensitivity analysis for unconditional Spearman's $\rho$ is to extend the neighboring relation to increment.

**Theorem 8.** *Spearman's $\rho$ is $\frac{30}{n}$-sensitive.*

*Proof.* When the neighboring datasets are defined by replacement, the proof is done in [15]. Now we prove that the sensitivity bound generalizes to neighboring datasets defined by increment. And we denote the incremented observation with $(a_{n+1}, b_{n+1})$. First, $\forall i \neq n+1$, $d_i$ changes at most 2. Thus $d_i^2 - (d_i - 2)^2 \leq 4(d_i - 1) \leq 4(m - 2)$, because $d_i$ is smaller than $m - 1$. Besides, $d_{n+1}$ is at most $m$. Therefore, the sensitivity of $\rho$ si bounded by

$$s(\rho) \leq \frac{30m(m-1)}{m(m^2-1)} \leq \frac{30}{m}$$

$\square$

**Definition 8** (Conditional Spearman's $\rho$). *We omit the pair indices $i, j$ and use $\rho_i$ to represent Spearman's $\rho$ in the $i$th block of the conditional variables. If there are $k$ blocks in total, then conditional Spearman's $\rho$ is defined as*

$$\rho = \frac{\sum_{i=1}^{k} w_i \rho_i}{\sqrt{\sum_{j=1}^{k} w_j}}$$

*where $w_i = n_i - 1$.*

**Theorem 9.** *Conditional Spearman's $\rho$ is $\frac{c_\rho \sqrt{k}}{\sqrt{n-k}}$-sensitive, where $c_\rho$ is an explicit constant typically close to 31.*

*Proof.* If the $i$th block contains $n_i$ observations, then $s(\rho_i) = \frac{30}{n_i}$.

Then we need to bound $\frac{w_i}{\sqrt{\sum_{j=1}^{k} w_j}}$ and its sensitivity.

$$\frac{w_i}{\sqrt{\sum_{j=1}^{k} w_j^2}} \leq \frac{n_i - 1}{\sqrt{n - k}}$$

and

$$s(\frac{w_i}{\sqrt{\sum_{j=1}^{k} w_j}}) \leq \frac{w_i'}{\sqrt{\sum_{j\neq i} w_j + w_i'}} - \frac{w_i}{\sqrt{\sum_{j=1}^{k} w_j}} \leq \frac{w_i' - w_i}{\sqrt{\sum_{j=1}^{k} w_j}} \leq \frac{1}{\sqrt{n-k}}$$

. Thus the complete sensitivity is bounded as follow.

$$s(\rho) \leq (\frac{w_i}{\sqrt{\sum_{j=1}^{k} w_j}} + s(\frac{w_i}{\sqrt{\sum_{j=1}^{k} w_j}}))(\rho_i + s(\rho_i)) - \frac{w_i}{\sqrt{\sum_{j=1}^{k} w_j}}\rho_i \leq \frac{31}{\sqrt{n-k}} + \frac{30}{c_1\sqrt{n-k}}$$

$\square$

## F Reconcile Sensitive Independence Test

As an attempt to reconcile independence tests with infinite sensitivity such as G-test or $\chi^2$-test in `Priv-PC`, we use subsample-and-aggregate and median aggregation with local sensitivity to stabilize these independence tests.

**Definition 9** (Subsample-and-aggregate [7]). *Let $f$ be the function of interest. In subsample-and-aggregate, the input database is randomly partitioned into $m$ blocks and $f$ is computed exactly on each block. The outcomes are then aggregated using a differentially private aggregation mechanism such as trimmed mean.*

In order to use subsample-and-aggregate, we clip the outcome of the independence test to a bounded range and estimate the median by adding noise calibrated to the smooth sensitivity [21].

**Definition 10** (Median Aggregation with Local Sensitivity [21]). *Let $S_{med}(x)$ represent the smooth sensitivity of the median of a given input $x$. $S_{med}(x)$ can be $\beta$ upper bounded by the following formula in $\mathcal{O}(n \log(n))$.*

$$S_{med}(x) = \max_{k=0,\cdots,n} (e^{-k\epsilon} \cdot \max_{t=0,\cdots,k+1} (x_{m+t} - x_{m+t-k-1}))$$

*where $m$ is the median index. Let $Z$ be a random value taken from an $(\alpha, \beta)$-admissible noise probability density function, then $med(x) + \frac{S_{med}(x)}{\alpha} \cdot Z$ is $(\epsilon, \delta)$-differentially private where $\epsilon$ and $\delta$ depends on $\alpha$ and $\beta$. For instance, the Laplace distribution $p(z) = \frac{1}{2} \cdot e^{-|z|}$ is $(\epsilon/2, \epsilon \ln(1/\delta)/2)$-admissible; the Gaussian distribution $p(z) = \frac{1}{2\pi} \cdot e^{-z^2/2}$ is $(\epsilon/\sqrt{\ln(1/\delta)}, \epsilon/2 \ln(1/\delta))$-admissible.*

---

**Algorithm 6:** Reconciled Independence Tests.

**Input:** $D$: dataset, $m$: number of blocks, $f$: independence test function, $T$: threshold, $\epsilon, \delta$: privacy parameters, $Z$: $(\alpha, \beta)$-admissible distribution.

1 **Function** *ReconciledIDT($D, f, T, \epsilon, \delta, Z$)*:
2      Partition the dataset in $m$ blocks $D_1, \cdots, D_m$.
3      Compute $f(D_1), \cdots, f(D_m)$.
4      Let $z \leftarrow Z$.
5      Output $med(f(D_1), \cdots, f(D_m)) + \frac{S_{med}(f(D_1), \cdots, f(D_m))}{\alpha} \cdot z$

---