[Reviews · NeurIPS 2020]

Review 1

Summary and Contributions: This paper proposes a differentially private causal graph discovery algorithm. The basic idea is that by adding the Laplace noise in the conditional independent test, one can achieve the differential with a certain privacy budget. The contribution of this paper is to improve the PC-EM algorithm to make it more practical with regard to speed and performance.

Strengths: This work improves the PC-EM algorithm to make it more practical with regard to speed and performance.

Weaknesses: However, there are some concerns: 1. Does the $ mean subsample in Algorithm 1 (line 2)? Is it necessary to sub-sample the data? The choose of 2. Is this method can only be applied in discrete data? If not why only use the discrete type of data in experiments. 3. It is also interesting to see how the performance of Priv-PC in a larger graph compare to the standard PC algorithm. 4. I'm still unclear about how serious the privacy problem is in PC. The example given in the second paragraph seems not enough to indicate the disease in a patient, because there could be several ways in PC to add a causal relationship. For example, consider the ground truth v-structure A->B<-C, and the current learned causal structure is $A-B-C; A-C$, and it is possible that if the newly added patent contains information "A is independent of C", and we can delete the edge A-C, then a v-structure will be detected. As a result, A->B<-C is added. However, this causal relationship does not reveal the patient's privacy since it only contains the information about "A is independent of C". ------- It is glad to see that the author adds some new experiments, but the authors did not respond to my concern about the privacy issues in the PC algorithm which might need further explanation. I will consider this paper as borderline and keep my score unchanged.

Correctness: Yes, they are correct.

Clarity: This paper is well written and well orgranize.

Relation to Prior Work: Yes, it has disccussed the related work but not very in detail about the difference bettween PC-EM and the proposed Priv-PC.

Reproducibility: Yes

Additional Feedback:


Review 2

Summary and Contributions: This paper studies the differentially private PC algorithm for constructing private preserving causal graph from data. It develops an improve version of the sparse vector technique (SVT) to achieve high accuracy and short running time. The general idea is that, when conducting independence tests in the PC algorithm, it first performs SVT on a small sampled subset with tweaked threshold for reducing type I errors, and then run independence tests again on the complete dataset for pairs that pass the filter. Experiments on four datasets show that the proposed method outperforms the state-of-the-art in terms both accuracy and running speed.

Strengths: It is an interesting idea to first filter out pairs that are below the threshold using a small sample. It is also a common and useful strategy to lower the threshold in order to reduce type I errors due to sampling. The paper provides theoretical analysis to the differential privacy, error bound, and sensitivity. The improvement in terms of the running speed is dramatic.

Weaknesses: The paper does not explain why SVT suffers from low accuracy. As a result, although the paper tries its best to explain the proposed method (which improves SVT) gives less running time and better utility, it is still not very clear. Some statements are not very accurate. For example, “It is unclear whether the differential privacy still holds given this compromise.” The greedy search compromise the optimal solution, which may compromise utility. The privacy guarantee still holds as proven. EM-PC achieves strict (epsilon,0)-DP. Priv-PC achieves (epsilon,delta)-DP as shown in Algorithm 2. Their results in the experiments are not comparable. It is unclear how much of the improvement of Priv-PC over EM-PC is due to the relaxation of privacy (a non-zero broken probability delta). Smaller privacy budget (<=1) is typically more meaningful in real world application, but the paper shows that EM-PC achieves better utility under small privacy budget. The paper uses vague language in technical sections (it is OK in the abstract or introduction), which is not preferred in a decent paper. For example, in Section 3.1, the PC algorithm uses independence tests that are “too many to obtain an acceptable privacy guarantee” EM-PC runs “a large number of independence tests” “only a few independence tests in causal discovery yield values above the threshold” The exact meanings of “too many, a large number of, a few” are vague.

Correctness: Some claims are not very accurate as pointed out above. The theoretical analysis is plausible.

Clarity: Yes.

Relation to Prior Work: Yes.

Reproducibility: Yes

Additional Feedback: ================================= EDIT AFTER AUTHOR RESPONSE ================================= I think the responses generally address my concerns, although I still think it would be better if they also show the difference in performances between epsilon-DP and epsilon,delta-DP for both methods, if possible. I would like to raise my score to 6.


Review 3

Summary and Contributions: The paper proposes a differentially private causal graph discovery algorithm, Priv-PC. Compared with another DP causal graph discovery algorithm based on exponential algorithm (called as EM-PC), the proposed Priv-PC algorithm adopts a paradigm called sieve-and-examine, which essentially applies the sub-sampled sparse vector technique and the Laplace mechanism in the two sub-processes.

Strengths: + How to construct DP causal graphs is important but under-explored. The only previous work, EM-PC, has limitations as pointed out by this paper: slow computation due to many independence tests and the intense calculation of the score function used in EM, and low utility due to the change of intrinsic workflow of PC. So this paper has its merits by presenting another practical (and better) algorithm. + I tend to agree with the authors that the proposed sieve-and-examine paradigm, which alternately executes sub-sampled sparse vector technique and output perturbation, can address low accuracy problem in the direct application of sparse vector technique. I think this is another main contribution of this work. + The work is relatively complete and its supplemental file includes the results of sensitivity analysis for several conditional independence tests, e.g., conditional Kenall's \tau and conditional Spearman's \pho, although the sensitivity analysis is not really challenging.

Weaknesses: - Somehow, I felt the methodological contribution is not high although the sieve-and-examine paradigm seems feasible in this context and the error bounds of type I error and type II error are not trivial. - Comparisons with the existing work (EM-PC) could be more fairly conducted. Recall the slow computation is because 1) involving too many independence tests in identifying the independence node set using EM; and 2) the intensive computation of the utility function used in EM. For 1), it would be better to calculate and report in the experiment the number of independence tests incurred in both EM-PC and Priv-PC. Note that EM-PC already reduces the number of conditional independence tests as it tries to identify conditional independence node set. In other words, "EM-PC makes the decision of edge elimination at each round of determining the conditional independence set for each node instead of making the decision by applying conditional independence test for each edge". So without reporting the numbers from experiments, the claim may not be right. Second, the current Priv-PC intentionally chooses Kendall's \tau for the independence test because the sensitivity values of other test metrics are high. On the contrary, EM-PC can support any test metrics because of the adopted score function (although whose computation is intense). Moreover, the proposed Priv-PC has much worse performance than EM-PC when privacy threshold \epsilon is low. The large \epsilon values basically cannot provide meaningful privacy protection.

Correctness: The theoretical results and the claims of the proposed method seem correct. The empirical methodology is also correct. The claims related to comparisons with the previous EM-PC need to be examined. See the comments in the weakness section.

Clarity: I think the paper is well written and can be easily followed. A minor comment. Somehow, I do not like the authors use the term of Priv-PC for their proposed method and call the previous work as EM-PC. Note that the original paper [30] named their method as PrivPC. I understand it is fine to rename the previous method as EM-PC for clearly comparison, but it would be better to name the proposed method as SE-PC (based on the sieve-and-examine paradigm).

Relation to Prior Work: See the second comment in the weakness section. [30] also presented PrivPC* to deal with numerical attributes when the data satisfy multi-Gaussian distribution.

Reproducibility: Yes

Additional Feedback: The number of edges in four chosen datasets is kind of small. In other words, the evaluation is only focused on very sparse causal graphs. The authors may check [30] to run simulation studies with different numbers of nodes, edges, and sparse ratio values. Regarding the rebuttal, I very appreciate the authors' efforts reporting the comparison between EM-PC and Priv-PC in terms of the number of independence tests, as shown in Table d in the rebuttal. The comparison results help understand the performance comparison in terms of execution time. However, the Priv-PC can only accommodate two independence test metrics, which I think is a big limitation. The comment on the exponential complexity of the brutal-force search of the accurate utility score in Priv-PC (as an advantage over EM-PC) is a bit misleading as Priv-PC does not have to use the optimal utility score. As discussed in my review, how to construct DP causal graphs is important and under-explored and the proposed sieve-and-examine paradigm is feasible to address low accuracy in the direct application of sparse vector technique. However, the methodological contribution seems below the bar of NeurIPS and overall I think the paper is a borderline one.


Review 4

Summary and Contributions: A differentially private causal graph discovery algorithm is proposed in this paper. A testing condition is derived to filter out variable pairs that are unlikely to be independent and hence fewer independence tests is needed. This would result in a reduction of computational cost. The whole algorithm incorporating this process in the PC method is presented. Experimental results are shown to compare the running time and accuracy of the method.

Strengths: There are very few works about the differential privacy issue of causal discovery. To the best of my knowledge, this paper could be one of the prior work in this area, and it is one of the areas that needs exploration. The paper presents an approach that is able to run faster than the only related existing approach, the EM-PC method. The work is not just empirical but also supported by theoretical proofs.

Weaknesses: The work focuses on the reduction of independent tests. Some of the traditional and non-differentially private algorithms related to PC algorithms are also addressing this problem of reducing the number of independent tests. While it is important to have a fast algorithm, but the speed of the algorithm should not be the main objective. But rather, the way that we can preserve the correctness of the causal algorithm without disclosing the information of the data should be focused on. The example used in the paper contains 100K samples and it is hard to imagine if any private information can be inferred. Datasets with 100K samples are pretty large for networks with few nodes and edges. Is the performance affected if smaller sample size is used? The authors in the feedback mentioned standard PC algorithm is used as the baseline for F1 score, but the paper says that F1 is measured based on the ground truth. It is known that PC algorithm may not have 100% accuracy especially with small sample size. So the authors need to clarify how F1 is measured in the experiments.

Correctness: Table 2 compares the running time of the algorithms. The running time should depend on a number of factors such as the parameters used. However, this table does not indicate the settings used. It is better to include the sensitives of the parameters too.

Clarity: Some discussions in the paper are relatively brief and hence readers may find it hard to understand. It is better to include more details about the algorithms and also the impact of the threshold, epsilon and subset size on the performance of the approach.

Relation to Prior Work: The authors are recommended to include a brief overview of the EM-PC in the paper so that it is easier for readers to have a comprehensive comparison between the works.

Reproducibility: Yes

Additional Feedback:

[Author Response · NeurIPS 2020]

We thank reviewers for their constructive comments, please see below for our response.

**Reviewer#1-1-Why only discrete data?** Priv-PC can only deal with discrete data because PC algorithm itself can
only deal with discrete data. We will make this clear in the revised version.

**Reviewer#1-2-Performance on larger graphs.** Thanks for the constructive feedback! Following the advice, we
evaluated Priv-PC on three larger causal graphs: (1) Alarm with 37 nodes and 46 edges (Figure 1a); (2) Child with 20
nodes and 25 edges (Figure 1b); (3) Sachs with 11 nodes and 17 edges (Figure 1c). We use standard PC algorithm as the
baseline (*i.e.* F1 score equaling 1 means the same performance as standard PC) and the results are consistent with the
evaluation on small graphs in the paper. We are also running EM-PC on these datasets but have not been able to collect
the results due to its long running time (about 10 hours per datapoint). We will include the new results in the revision.

| #idp tests | Priv-PC | EM-PC |
|---|---|---|
| Asia | 95 | 216 |
| Cancer | 37 | 57 |
| Earthquake | 40 | 61 |
| Survey | 29 | 38 |
| Alarm | 1843 | 12979 |
| Child | 1162 | 7393 |
| Sachs | 165 | 1224 |

(a) Alarm.　　(b) Child.　　(c) Sachs.

(d) The number of indenpendence tests in Priv-PC and EM-PC.

**Reviewer#2-1-Why SVT suffers from low accuracy.** We refer to [4] as an explanation for the low accuracy of SVT.

**Reviewer#2-2-Influence of greedy search on the privacy proof.** Thanks for the review but there seems to be a
misunderstanding about the statement due to our insufficient elaboration. Given the greedy search compromise, EM-
PC's original privacy guarantee might not hold because the sensitivity of the utility score calculated with greedy search
is not necessarily the same as the one used in the privacy proof. To validate the concern, we use greedy search to
calculate the utility scores on neighboring datasets and observe a difference of 3, larger than the claimed sensitivity 1
used in the proof. We will make the statement more clear in the revision.

**Reviewer#2-3-Is improvement due to relaxation of privacy?** We appreciate the reviewer's feedback but there
seems to be a misunderstanding about EM-PC due to our insufficient elaboration. EM-PC is $(\epsilon, \delta)$-differentially private
because it is composed of multiple EMs with advanced composition [3]. The composition process makes EM-PC
$(\epsilon, \delta)$-differentially private although exponential mechanism itself follows $\epsilon$-differential privacy. Thus in our evaluation,
EM-PC and Priv-PC are compared under exactly the same privacy guarantee. We will make it more clear in the revision.

**Reviewer#2-4-Performance in the high privacy region.** We agree that smaller privacy budget ($\leq 1$) provides
stronger privacy guarantee, but we argue that in many data-intense tasks such as deep learning, privacy budget larger
than 1 is acceptable and even a common case. For example, in Abadi et al.'s pioneering work [1], they use $\epsilon = 2, 4, 8$ in
their experiments for a neural network with one hidden layer. Other examples of large $\epsilon$ can be found in many well-cited
papers. Due to space limitation, we list two here [2, 5]. Priv-PC typically outperforms EM-PC somewhere between 2
and 8, which is an acceptable and practical privacy regime in many real-world applications.

**Reviewer#3-1-Record the number of independence tests in EM-PC and Priv-PC.** Thanks for the advice! As
suggested, we have recorded the number of independence tests needed in Priv-PC and EM-PC on the 4 original datasets
and the 3 new datasets as shown in Table 1d. The results show that Priv-PC saves 24%~56% independence tests on
small graphs and 94%~97% on larger graphs compared to EM-PC. We will include the new results in the revision.

**Reviewer#3-2-Compatibility with different independence tests.** We admit that Priv-PC can only accommodate
Kendall's $\tau$ and Spearman's $\rho$ currently and the reconciling of other independence tests is an interesting future direction.
On the other hand, although theoretically EM-PC can leverage any independence test, the only known way to obtain
the accurate utility score is brutal-force search with exponential complexity, which is almost impossible to implement.
Thus, we view Priv-PC as a step forward compared to EM-PC since it is implementable.

**Reviewer#4-1-Is speed a big issue?** We entirely agree that for most DP algorithms, the privacy-utility trade-off
is the biggest challenge. We emphasize the speedup of Priv-PC because the only prior work, EM-PC, suffers from
extremely slow computation. The slowing down stems from the privacy augmentation, which cannot be fully addressed
by traditional methods developed for PC algorithm. Thus, we would like to find a way to achieve an elegant balance
between speed, utility and privacy guarantee.

[1] M. Abadi, A. Chu, I. Goodfellow, H. B. McMahan, I. Mironov, K. Talwar, and L. Zhang. Deep learning with differential privacy.
[2] N. Agarwal, A. T. Suresh, F. X. X. Yu, S. Kumar, and B. McMahan. cpsgd: Communication-efficient and differentially-private
distributed sgd.
[3] C. Dwork, G. N. Rothblum, and S. Vadhan. Boosting and differential privacy.
[4] M. Lyu, D. Su, and N. Li. Understanding the sparse vector technique for differential privacy.
[5] Y.-X. Wang, B. Balle, and S. P. Kasiviswanathan. Subsampled rényi differential privacy and analytical moments accountant.


[Meta-Review · NeurIPS 2020]

Although this paper is somewhat incremental, it nevertheless offers a significant practical improvement on the state-of-the-art for private causal graph discovery, and after discussion and consideration of the author feedback, I believe that it meets the bar for acceptance. Authors, please be sure to follow through on the improvements promised in your feedback for the camera-ready version.